# Unveiling the Role of the Work Environment in the Quality of Life of Menopausal Physicians and Nurses

**DOI:** 10.3390/ijerph20186744

**Published:** 2023-09-12

**Authors:** Gauri Bapayeva, Milan Terzic, Yuliya Semenova, Antonio Sarria-Santamera, Arnur Gusmanov, Gulzhanat Aimagambetova, Antonio Simone Laganà, Vito Chiantera, Nazira Kadroldinova, Talshyn Ukybassova, Kuralay Kongrtay, Meruyert Abdukassimova, Karlygash Togyzbayeva, Sanja Terzic

**Affiliations:** 1Clinical Academic Department of Women’s Health, CF “University Medical Center”, Turan Ave. 32, Astana 010000, Kazakhstan; gauri.bapaeva@umc.org.kz (G.B.); milan.terzic@nu.edu.kz (M.T.); talshynu@yandex.ru (T.U.); togyzbaevaka@mail.ru (K.T.); 2Department of Surgery, School of Medicine, Nazarbayev University, Zhanybek-Kerey Khans Street 5/1, Astana 010000, Kazakhstan; yuliya.semenova@nu.edu.kz (Y.S.); gulzhanat.aimagambetova@nu.edu.kz (G.A.); nazira.kadroldinova@nu.edu.kz (N.K.); kkongrtay@nu.edu.kz (K.K.); meruyert.abdukassimova@nu.edu.kz (M.A.); 3Department of Obstetrics, Gynecology and Reproductive Sciences, University of Pittsburgh School of Medicine, 300 Halket Street, Pittsburgh, PA 15213, USA; 4Department of Biomedical Sciences, School of Medicine, Nazarbayev University, Zhanybek-Kerey Khans Street 5/1, Astana 020000, Kazakhstan; antonio.sarria@nu.edu.kz; 5Department of Health Promotion, Mother and Child Care, Internal Medicine and Medical Specialties (PROMISE), University of Palermo, 90133 Palermo, Italy; antoniosimone.lagana@unipa.it (A.S.L.); vito.chiantera@unipa.it (V.C.); 6Unit of Gynecologic Oncology, National Cancer Institute, IRCCS, Fondazione “G. Pascale”, 80131 Naples, Italy; 7Department of Medicine, School of Medicine, Nazarbayev University, Zhanybek-Kerey Khans Street 5/1, Astana 010000, Kazakhstan; sanja.terzic@nu.edu.kz

**Keywords:** menopause, menopausal quality of life, physicians, nurses, clinical setting

## Abstract

Background: There is a lack of scientific evidence regarding the specific challenges faced by menopausal medical professionals in different work settings. This study aims to investigate the relationship between work environment and the menopausal quality of life (QoL) in physicians and nurses. Methods: This survey was conducted using the Menopausal Quality of Life Questionnaire (MENQOL) with a sample of 35 menopausal physicians and 95 nurses employed in health facilities in Astana and Kyzylorda cities, Kazakhstan. Results: Physicians reported a higher frequency of menopausal symptoms compared to nurses. The difference was statistically significant (*p* < 0.05) for symptoms such as decreased productivity (60.00% vs. 38.20%), flatulence or gas pains (71.43% vs. 48.39%), weight gain (79.41% vs. 61.80%), changes in skin appearance (79.59% vs. 50.00%), and changes in sexual desire (58.82% vs. 33.70%). Physicians with managerial duties had a significantly higher occurrence of vasomotor symptoms compared to non-managerial physicians (mean 3.35 ± 2.14 vs. 1.69 ± 0.89) and also had a higher mean psychological score (mean 3.26 ± 1.28 vs. 2.29 ± 1.19). Conclusions: These findings reflect differences between the menopause effects related to work environment for doctors and nurses, and shed light on the specific challenges faced by them during menopause. In addition, it is important to consider socio-demographic and workplace-related factors in investigating their impact on the QoL.

## 1. Introduction

Menopause is a natural decline in follicular function of the ovaries, accompanied by a decrease in serum estrogen levels. Currently, a woman is considered menopausal if she has no menstrual periods for 12 consecutive months [1]. The prevalence of menopausal symptoms varies across regions, with approximately 74% of women in Europe, 36–50% of women in North America, and 45–69% of women in South America experiencing these symptoms. However, it is estimated that menopausal symptoms are less common in Asia (22–63%) [2]. The onset of menopause can vary greatly among individuals, occurring anywhere from the mid-thirties to the mid-fifties. With increasing global life expectancy, more women are experiencing menopause in the modern era. It is projected that by 2030, the global menopausal population will reach 1.2 billion women, with 47 million new cases annually [3]. The onset and severity of menopausal symptoms are influenced by various factors, including genetics, body weight, habits such as smoking and alcohol addiction, diet and nutrient intake, and the duration of menstrual periods [4,5]. The most common symptoms include vasomotor reactions such as hot flashes and/or night sweats. In addition, women may experience anxiety, depression, fatigue, insomnia, musculoskeletal pain, vaginal dryness, and sexual dysfunction [6].

The quality of life (QoL) of menopausal women can experience a significant decline, which can be attributed to both the symptoms caused by menopause and the associated health conditions, including urinary incontinence, osteoporosis, cardiovascular disease, and gynecological disorders [7]. These symptoms can be debilitating and have a profound negative impact on various aspects of a woman’s life, particularly because they occur during a life stage when women play important roles in society, family, and work. However, in many communities worldwide, it may not be socially acceptable to openly discuss these symptoms or seek professional help, making it challenging to accurately assess the magnitude of the problem. Therefore, it would be beneficial to investigate menopause-related QoL using specifically developed tools. To facilitate this task, several questionnaires have been proposed for use in clinical settings [8,9].

Kazakhstan is one of the Central Asian countries. After the Soviet Union collapse and subsequent economic recession, the Kazakhstani healthcare system passed through profound reforms. Notably, only around 3% of gross domestic product (GDP) is allocated for the healthcare system, while the average expenditures in the post-Soviet neighboring countries are around 6.5% [10,11]. Government priorities in this sector include developing the country’s primary healthcare networks, improving its public health administration system, providing expanded medical personnel training, enhancing mother and child health services, and emphasizing preventive measures such as diagnostics, treatment of lifestyle diseases, patient rehabilitation, and development of personalized medicine. The Kazakhstani government has approved a budget of USD 7.5 billion for the State Program for the Development of Healthcare 2020–2025 [12]. According to the National Agency for Statistics, as of 1 January 2023, more than 260,200 medical workers, including 76,000 doctors, and 186,000 nurses, work in the Kazakhstani healthcare system [13].

The symptoms associated with menopause can significantly affect a woman’s professional life, resulting in reduced work productivity, increased absenteeism due to sick leave, and potential early retirement. These consequences impose both indirect costs on employers and direct costs on the healthcare system. Women working in professions that demand high levels of productivity may struggle to maintain their performance and may consider transitioning to less demanding careers [14]. Among various professions, the medical field places particular demands on individuals, requiring skills such as concentration, multitasking, empathy, and emotional regulation [15]. Consequently, it is reasonable to hypothesize that healthcare professionals going through menopause may encounter difficulties in managing their symptoms. While a limited number of studies have explored the specific challenges faced by menopausal nurses in various work environments, there is a lack of research on female physicians, despite the need for such data to guide the development of targeted interventions. Therefore, this study aims to investigate the relationship between work environment and the menopausal quality of life (QoL) in physicians and nurses. We hypothesize that physicians and nurses possess varying QoL scores due to differences in their associated work-related characteristics.

## 2. Materials and Methods

### 2.1. Data Collection and Participants

A cross-sectional study using the Menopausal Quality of Life Questionnaire (MENQOL) was conducted with 130 menopausal physicians (*N* = 35) and nurses (*N* = 95). Between 2022 and 2023, a survey was conducted in four hospitals within the University Medical Center (UMC) in Astana and Kyzylorda, utilizing a self-administered questionnaire. Clinics involved in the study were the National Scientific Center for Mother and Child (NRCMC), the Republican Diagnostic Center (RDC), and the National Center for Children’s Rehabilitation (NCCR), Kazakhstan.

The study included persons who met the following inclusion criteria: (1) physicians and nurses from the aforementioned clinics and (2) aged 40 years and older. Respondents with severe health issues, including physical and mental diseases that hindered their ability to take the survey, were excluded from the study. Additionally, women receiving hormone replacement therapy were excluded as well.

The required sample size was calculated based on an anticipated effect size, a Type I error rate of 5%, and a power of 80%. The sample size was estimated to be 240 participants. However, due to non-response, we were able to obtain 130 eligible responses. Thus, the response rate accounted for 54%. An email containing an invitation to participate in the study, the informed consent form, and a questionnaire was sent to the participants’ corporate email addresses. Participation was entirely voluntary, and individuals had the freedom to discontinue the survey whenever they chose. Those who agreed to take part returned the completed questionnaire and a signed informed consent form to the researchers.

### 2.2. Study Instrument

A questionnaire including various items were utilized to assess socio-demographic and job-related variables. Physicians and nurses who took part in the study were required to complete a self-reported questionnaire. The main focus of interest was the quality of life during menopause, which was assessed using the MENQOL inventory. The survey instrument consisted of three sections, with the initial section gathering demographic information such as age, marital status, and educational level. The second section encompassed the MENQOL questionnaire, providing a comprehensive evaluation of menopause-related symptoms and their impact on various aspects of an individual’s quality of life. The third section had several questions aimed to evaluate different aspects of their employment, such as job position (physician or nurse), work-related characteristics (such as weekly work hours and night shifts), job satisfaction, and department affiliation. Participants’ menopause status was self-reported, with participants asked to indicate their menopause status. Based on their responses, three categories were established and analyzed: pre-menopause (a period in women’s life preceding the onset of menopause), perimenopause (the transitional phase between the fertile period and overt menopause), and postmenopause (defined as 12 months of complete amenorrhea in women older than 45 years).

The MENQOL enabled assessment of quality of life related to various groups of menopausal symptoms: (1) physical with 16 items, (2) psychological including 7 questions, (3) vasomotor with 3 items and (4) sexual domain, which comprised 4 questions. The mentioned physical, psychological, vasomotor, and sexual domains were assessed using 16, 7, 3, and 4 items of the MENQOL questionnaire [16]. Past studies established that MENQOL possesses high validity and reliability in various populations [17,18]. The calculation of the MENQOL score for each participant required coding the responses for each question in Likert scale ranging from 1 to 8. The value of “one” indicated that a participant did not have a specific symptom, while “two” or more corresponded to the case when a women experienced a symptom. The range from 2 to 8 corresponded to the extent of being bothered with a particular menopausal symptom ranging from “not bothered at all” to “extremely bothered”.

The questionnaire underwent cross-cultural adaptation and translation for use in the study settings. The translation process involved professional trilingual (English–Russian–Kazakh) interpreters conducting both forward and backward translations into Kazakh and Russian languages. Discrepancies in translations and interpretations were resolved through comprehensive discussions by a panel of experts. The Kazakh and Russian translated versions of the survey were pretested on 20 participants to evaluate the clarity and appropriateness of the translated items and responses. In this study, the Cronbach α values were 0.91 for vasomotor domain, 0.95 for psychological domain, 0.95 for physical domain, and 0.9 for sexual domain.

### 2.3. Statistical Analysis

In the descriptive analysis, categorical variables were summarized using frequencies and proportions, while mean and standard deviations were utilized in the univariate analysis of continuous variables. The scores for the MENQOL items were averaged for each participant and total MENQOL and domain-specific scores were calculated. Both continuous MENQOL scores and the proportion of participants with specific symptoms were used in univariable and bivariate analyses. Bivariate analysis aimed to compare MENQOL scores between physicians and nurses using the independent two-sample *t*-test. Further, investigation of associated factors with QoL was performed separately for doctors and nurses to see whether they have different set of associated characteristics. The one-way analysis of variance (ANOVA) was in use when there were three categories in an independent variable, while the cases with two categories required the usage of the independent two sample *t*-test. Assumptions for these parametric tests, such as normal distribution and homoscedasticity, were evaluated by visual aids (histograms and QQ-plots), the Shapiro–Wilk test, and Levene’s test. Violation of assumptions dictated the utilization of non-parametric alternative tests (Mann–Whitney U test and Kruskal–Wallis test). Post hoc pairwise comparisons with the Bonferroni correction were performed after conducting ANOVA and Dunn’s test after Kruskal–Wallis test. Stata software (StataCorp, College Station, TX, USA, 2017) was used for the statistical analysis in this study.

## 3. Results

Table 1 presents a descriptive analysis of the characteristics of the participants. The majority of both physicians (65.7%) and nurses (51.6%) were aged 50 years or older. A significantly higher proportion of physicians (65.7%) reported being married compared to nurses (43.2%). There were statistically significant differences in the education levels of physicians and nurses (*p* < 0.001). Specifically, the majority of doctors (82.9%) had higher education, while many nurses had completed specialized secondary education (71.6%). Regarding employment, similar proportions of doctors and nurses worked at the National Scientific Center for Mother and Child (39.4% and 47.2%, respectively) and the Republican Diagnostic Center (42.4% and 40.6%, respectively). A majority of physicians (58.1%) worked in the internal medicine department, with only a small percentage working in the surgical department (12.9%). On the other hand, around one-third of nurses were employed in surgical (37.7%), internal medicine (34.8%), or other departments (27.5%). A significantly higher proportion of nurses (55.3%) had night shifts compared to physicians (17.1%). In contrast, while half of the doctors indicated having managerial responsibilities, only a quarter of nurses (22.8%) reported having management duties. No significant differences were observed between physicians and nurses regarding their general satisfaction with their job and various aspects of employment such as salary and work environment. In terms of menopausal status, almost half of the respondents among physicians (48.6%) and nurses (43.7%) identified themselves as being postmenopausal, while 34.3% of doctors and 36.8% of nurses indicated being in the premenopausal period. The smallest percentages of physicians (17.1%) and nurses (19.5%) reported being in the perimenopausal stage.

Analysis of the prevalence of specific menopausal symptoms is presented in Table 2. It was found that one-third of physicians and nurses experienced at least one menopausal symptom. Physical symptoms were the most commonly reported symptoms among both physicians and nurses. Compared to nurses, a significantly higher proportion of physicians reported experiencing menopausal symptoms such as accomplishing less than they used to (60% vs. 38.2%), flatulence or gas pains (85.3% vs. 69.4%), weight gain (79.4% vs. 50%), change in appearance, texture, or tone of skin (79.6% vs. 50%), and change in sexual desire (58.8% vs. 33.7%). No menopausal symptoms were reported significantly more by nurses than physicians. Analysis of the variation of symptoms between different age categories is presented in Appendix A Appendix A. The average total MENQOL score was 3.12 (SD 1.65), with average scores of 2.58 (SD 2.09) for vasomotor symptoms, 2.73 (SD 1.92) for psychological symptoms, 3.36 (SD 1.67) for physical symptoms, and 2.77 (SD 2.16) for sexual symptoms. There were no statistically significant differences between physicians and nurses in the mean values of vasomotor (means 2.43 vs. 2.63), psychological (2.73 vs. 2.73), physical (3.51 vs. 3.29), sexual (2.79 vs. 2.76), and all domains (3.13 vs. 3.11) (Figure 1, Figure 2, Figure 3, Figure 4 and Figure 5).

Examining the associations between socio-demographic and work-related characteristics and different domains of QoL among physicians and nurses revealed a significant association between vasomotor scores and having management duties for physicians (*p* = 0.019) (Table 3). Specifically, doctors with managerial responsibilities were significantly more bothered by vasomotor symptoms (mean 3.35 ± 2.14) compared to doctors without such duties (mean 1.69 ± 0.89). Age and the number of work hours per week were significantly associated with vasomotor scores among nurses. Post hoc pairwise comparison using Dunn’s test showed that nurses aged between 50 and 54 years had significantly higher vasomotor scores (mean 3.51 ± 2.27) compared to both the “40–49” age group (mean 2.24 ± 2.09) and those aged older than 54 years (mean 2.12 ± 1.9). In addition, nurses working less than 40 h per week had significantly higher mean vasomotor scores (3.68 ± 2.47) compared to those working from 40 to 60 h (mean 2.14 ± 1.82, *p* = 0.004).

Physicians with management duties had a significantly higher mean psychological score compared to their colleagues without managerial responsibilities (mean 3.26 ± 1.28 vs. 2.29 ± 1.19, *p* = 0.045). Nurses who were dissatisfied with their job overall were significantly more bothered by psychological (mean 3.69 ± 2.41 vs. 2.23 ± 1.75, *p* = 0.002), physical (mean 4.0 ± 1.97 vs. 2.92 ± 1.57, *p* = 0.009), and sexual (mean 3.5 ± 2.56 vs. 2.37 ± 2.0, *p* = 0.027) menopausal symptoms compared to satisfied participants. Furthermore, satisfaction with the work environment was associated with better quality of life related to psychological symptoms among nurses (*p* = 0.012). Similarly, nurses who were satisfied with their salary were significantly less bothered by physical symptoms of menopause compared to dissatisfied nurses (mean 3.61 ± 1.91 vs. 2.65 ± 1.36, *p* = 0.033). Self-reported menopause status was only significantly associated with sexual scores among nurses, with significantly increased mean scores from 2.09 to 3.89 for nurses in the premenopause to postmenopause stages.

## 4. Discussion

The findings of our study suggest that the work environment influences the well-being of nurses and physicians, and therefore provide insights into the specific challenges experienced by healthcare providers during menopause and emphasize the importance of considering various socio-demographic and workplace-related factors when understanding their impact on quality of life. In this study, it was important to address this study question among healthcare workers because healthcare practitioners’ perceptions of their health and well-being could potentially be shaped by their interactions with patients and the conditions of their work environment [19].

Previous studies showed that the prevalence of menopausal symptoms varies worldwide [20,21]. In our study, we found that physical symptoms were the most common among both physicians and nurses, while earlier studies among Indian [20] and Australian [22] women showed that psychological symptoms are the leading symptoms among menopausal women. In contrast, our study demonstrated that psychological symptoms were not the leading concerns mentioned by healthcare workers independently of menopause status. Several studies support that the higher prevalence of physical symptoms than other dimensions can be explained by physiological changes occurring in the hormonal level of perimenopausal and postmenopausal women [21].

The results of this study also support that menopausal symptoms interrupt work performance and general quality of life among healthcare workers [23,24]. Although the previous studies showed that the well-being and work abilities of menopausal healthcare workers were mostly affected by psychological symptoms [23], our study showed that physical symptoms were the most problematic in this population. The other study conducted in China showed that both physical and psychological symptoms had the highest impact on QoL [25]. Considering that menopausal symptoms are correlated with each other [22], this variance in symptom prevalence and its influence on QoL between different studies can be acceptable.

However, while in the general population, the most influential factor on QoL was vasomotor symptoms, this is not the case among healthcare workers. This finding can be explained by the high levels of work-related stress and job burnout among healthcare workers, which lead to a more significant influence of psychological and physical symptoms on QoL than vasomotor and sexual symptoms [25]. However, our study also showed that age and working hours were associated with vasomotor symptoms among nurses. This result is consistent with the previous study conducted among Iranian women [26]. In addition, age can be directly correlated with psychological symptoms, which can be explained by the presence of more physical health concerns with increasing age and more anxiety about health problems [7]. In addition, previously it was found that vasomotor symptoms are associated with impaired work performance [27], whereas our study found the inverse relationship is also present and that increased working hours worsen vasomotor symptoms of menopause. As for the sexual domain, we found that menopausal status was significantly associated with the sexual aspect of quality of life among nurses.

One of the important findings we identified is that managerial responsibilities lead to greater distress from menopausal symptoms with lower psychological QoL. Although previous studies showed that there is no association between work stress and menopausal symptoms [24], in our study we emphasize that additional stress from managerial positions can have a significant on the quality of life among menopausal women. In general, there is inconsistency among various studies exploring the relationship between managerial responsibilities and menopausal symptoms, with some studies reporting positive effects while others reporting negative effects. For instance, High and Marcellino conducted a study on a cohort of American women and found that managers were less affected by both physical and psychological symptoms of menopause compared to non-managers. These differences were significant for symptoms such as weight gain, bloating, depression, irritability, and mood changes [28]. However, Matsuzaki et al. observed that nurses in managerial positions more frequently reported feeling unhappy or depressed compared to other nurses. Nevertheless, non-managerial nurses reported higher levels of physical overload, problems with job control, and workplace environment compared to managers [29]. Possibly the differences in how women in managerial positions express their menopausal symptoms are attributed to cultural variations between countries. Further research is needed to investigate the impact of managerial responsibilities on menopausal symptoms, especially as an increasing number of women of menopausal age enter or remain in the labor market, many of whom hold managerial positions.

Satisfaction with the overall job and specific aspects of it significantly influenced the psychological, physical, and sexual scores of quality of life among nurses. Our study showed that individuals who reported higher levels of satisfaction with their work environment and/or salary had fewer complaints regarding physical and psychological symptoms associated with menopause. This result supports the previous study by Márquez Membrive et al. that demonstrated better QoL among women with higher levels of job satisfaction [30]. A similar observation was reported in the study among the Serbian population [7]. According to that study, higher work income, which contributes to overall job satisfaction, was associated with improved psychological QoL [7]. In addition, social support in working organizations was shown to increase overall QoL among menopausal women [25]. Considering the results of the previous and the current study, the adverse effect of menopausal experience on work performance and work outcomes should be considered by workplaces, including healthcare institutions.

This study possesses several strengths and limitations. On one hand, it represents one of the first research efforts in Kazakhstan to examine the quality of life among menopausal physicians and nurses in relation to their socio-demographic and workplace characteristics. The utilization of a cross-sectional study design allowed for the investigation of multiple exposure statuses within a relatively short timeframe. By employing the well-established MENQOL instrument, the quality of life of healthcare providers was assessed with a high degree of validity and reliability. Moreover, since this questionnaire has been standardized and utilized in various settings globally, the findings from this study can be compared with the results of numerous other studies conducted worldwide.

However, it is crucial to acknowledge the limitations of this study. Firstly, the cross-sectional study design employed does not enable the establishment of a temporal relationship between the variables under investigation. As the data relied solely on participants’ self-reporting rather than objective documentation or instruments, the conclusions may be susceptible to various sources of biases. Furthermore, due to non-responses to certain questions in the survey instrument, a notable percentage of respondents who initially agreed to participate in the interview were excluded from the analysis. Potential systematic differences between participants whose responses were analyzed and those excluded could potentially impact the generalizability of the findings to the larger population. Additionally, the sample size is relatively small, which limited the ability to achieve sufficient statistical power and may affect the generalizability of the findings to a broader population.

## 5. Conclusions

The study aimed to determine the prevalence of menopausal symptoms and assess the quality of life, as well as the associated risk factors, among physicians and nurses in Kazakhstan. Our study’s results indicate that the work environment plays a role in influencing the well-being of nurses and physicians. As a result, these findings shed light on the unique difficulties that healthcare providers encounter during menopause and underscore the significance of examining diverse socio-demographic and workplace-related aspects to comprehend their influence on the quality of life. We found that several factors were identified as significant contributors to the domain-specific MENQOL scores among physicians and nurses. Our study suggests that work environment factors influence different nurses and physicians, and therefore provides insights into the specific challenges experienced by healthcare providers during menopause and emphasizes the importance of considering various socio-demographic and workplace-related factors when understanding their impact on quality of life. We suggest that leadership within healthcare organizations ought to recognize suitable interventions and take steps to introduce them, aiming to enhance the well-being of healthcare professionals.

## Figures and Tables

**Figure 1 ijerph-20-06744-f001:**
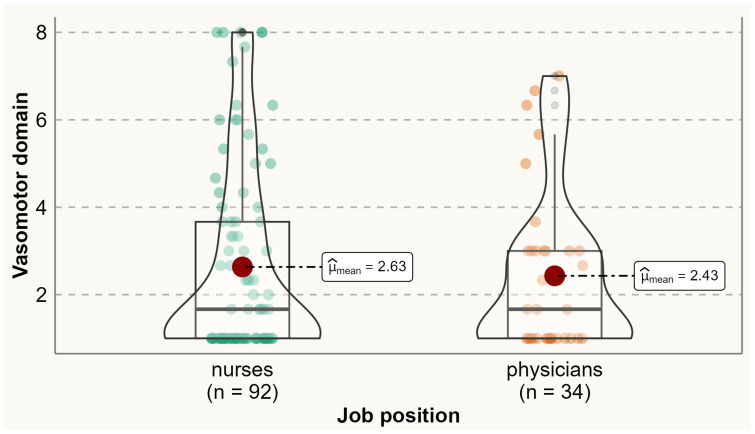
Comparing MENQOL vasomotor scores between physicians and nurses. Comparison of group means: $t_{Welch}\;\left(70.98\right)=0.53,\;p=0.60,\;{\hat{g}}_{Hedges}=0.10,\;CI_{95\%}\left[-0.27,\;0.47\right],\;\;n_{obs}=126$.

**Figure 2 ijerph-20-06744-f002:**
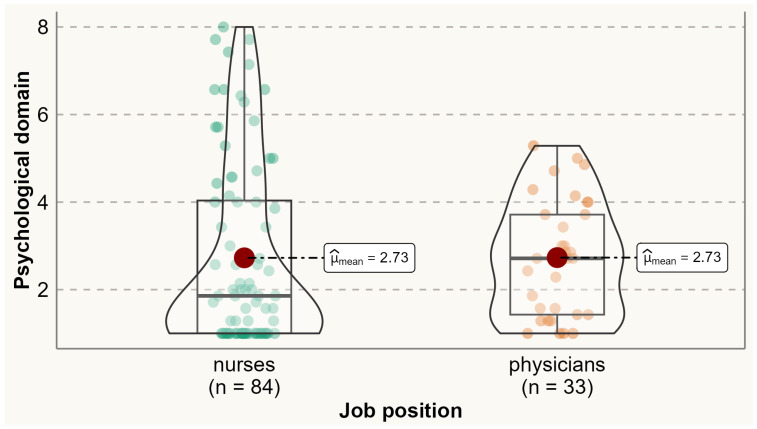
Comparing MENQOL psychological scores between physicians and nurses. Comparison of group means: $t_{Welch}\;\left(92.93\right)=-0.02,\;p=0.99,\;{\hat{g}}_{Hedges}=-3.04\;10^{-3},\;CI_{95\%}\left[-0.36,\;0.36\right],\;n_{obs}=117$.

**Figure 3 ijerph-20-06744-f003:**
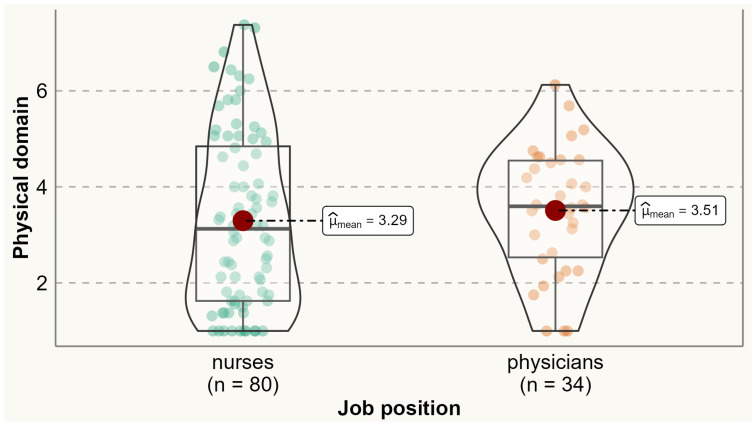
Comparing MENQOL physical scores between physicians and nurses. Comparison of group means: $t_{Welch}\;\left(83.92\right)=-0.70,\;p=0.48,\;{\hat{g}}_{Hedges}=-0.13,\;\;CI_{95\%}\left[-0.51,\;0.24\right],\;n_{obs}=114$.

**Figure 4 ijerph-20-06744-f004:**
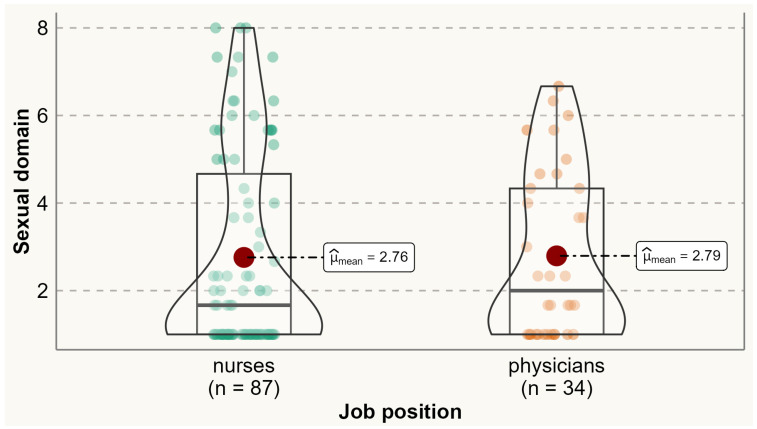
Comparing MENQOL sexual scores between physicians and nurse. Comparison of group means: $t_{Welch}\;\left(71.14\right)=-0.09,\;p=0.93,\;{\hat{g}}_{Hedges}=-0.02,\;CI_{95\%}\left[-0.39,\;0.36\right],\;n_{obs}=121$.

**Figure 5 ijerph-20-06744-f005:**
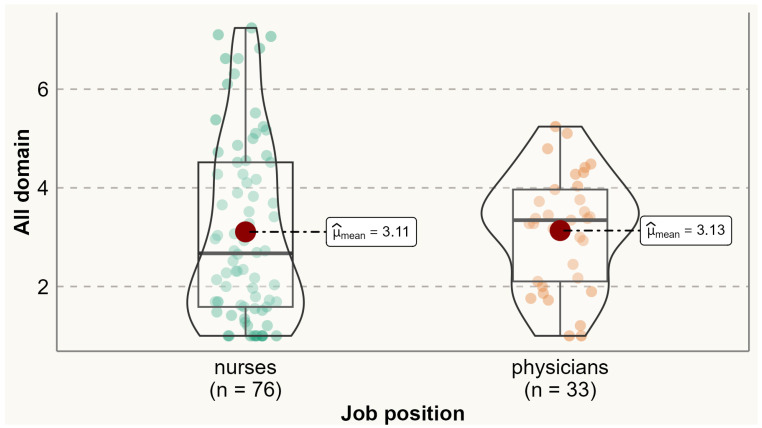
Comparing total MENQOL scores between physicians and nurses. Comparison of group means: $t_{Welch}\;\left(91.99\right)=-0.09,\;p=0.93,\;{\hat{g}}_{Hedges}=-0.02,\;CI_{95\%}\left[-0.39,\;0.35\right],\;n_{obs}=109$.

**Table 1 ijerph-20-06744-t001:** Descriptive analysis of socio-demographics and variables related to work for physicians and nurses.

Variables	Physicians (*N* = 35)	Nurses (*N* = 95)	*p*-Value
Age			0.152
40–49	12 (34.29%)	46 (48.42%)	
50–54	11 (31.43%)	31 (32.63%)	
55+	12 (34.29%)	18 (18.95%)	
Marital status, N (%)			0.023
Married	23 (65.71%)	41 (43.16%)	
Other (single, divorced, widowed)	12 (34.29%)	54 (56.84%)	
Education, N (%)			<0.001
Higher education	29 (82.86%)	24 (25.26%)	
Specialized secondary education	1 (2.86%)	68 (71.58%)	
Other *	5 (14.29%)	3 (3.16%)	
Medical organization, N (%)			0.434 **
National Scientific Center for Mother and Child	13 (39.39%)	42 (47.19%)	
Republican Diagnostic Center	14 (42.42%)	36 (40.45%)	
Republican Children’s Rehabilitation Center	4 (12.12%)	3 (3.37%)	
University Health Center	2 (6.06%)	7 (7.87%)	
Department, N (%)			0.024
Surgical	4 (12.9%)	26 (37.68%)	
Internal medicine	18 (58.06%)	24 (34.78%)	
Other (Master’s and Doctorate degrees)	9 (29.03%)	19 (27.54%)	
Work hours per week, N (%)			0.054
<40 h	6 (17.14%)	26 (27.66%)	
40–60 h	26 (74.29%)	48 (51.06%)	
>60 h	3 (8.57%)	20 (21.28%)	
Having night shifts, N (%)			<0.001
Yes	6 (17.14%)	52 (55.32%)	
No	29 (82.86%)	42 (44.68%)	
Having management responsibilities, N (%)			0.004
Yes	16 (50%)	21 (22.83%)	
No	16 (50%)	71 (77.17%)	
General satisfaction with job, N (%)			0.177
Satisfied	26 (76.47%)	58 (63.74%)	
Dissatisfied	8 (23.53%)	33 (36.26%)	
Satisfaction with salary, N (%)			0.246
Satisfied	15 (42.86%)	29 (31.87%)	
Dissatisfied	20 (57.14%)	62 (68.13%)	
Satisfaction with work environment, N (%)			0.248
Satisfied	25 (71.43%)	76 (80.85%)	
Dissatisfied	10 (28.57%)	18 (19.15%)	
Self-reported menopause status, N (%)			0.882
Premenopause	12 (34.29%)	32 (36.78%)	
Perimenopause	6 (17.14%)	17 (19.54%)	
Postmenopause	17 (48.57%)	38 (43.68%)	

* Other degrees, including Master’s and Doctorate. ** *p*-value is calculated using Fisher’s exact test.

**Table 2 ijerph-20-06744-t002:** Proportion of physicians and nurses with menopausal symptoms.

Symptom	Physicians	Nurses	*p*-Value
**Vasomotor Symptoms**			
Hot flashes	42.86%	41.94%	0.925
Nights sweats	38.24%	34.94%	0.661
Sweating	37.14%	37.23%	0.992
**Psychological symptoms**			
Being dissatisfied with their personal life	47.06%	32.26%	0.124
Feeling anxious or nervous	57.58%	44.57%	0.199
Experiencing poor memory	61.76%	47.25%	0.149
Accomplishing less than they used to	60.00%	38.20%	0.028
Feeling depressed, down or blue	50.00%	38.20%	0.235
Being impatient with other people	58.82%	44.32%	0.151
Feelings of wanting to be alone	51.52%	39.56%	0.234
**Physical symptoms**			
Flatulence or gas pains	71.43%	48.39%	0.02
Aches in the muscles and joints	64.71%	53.41%	0.259
Feeling tired or worn out	85.29%	69.41%	0.075
Difficulty sleeping	58.82%	52.22%	0.55
Aches in the back, neck or head	73.53%	59.55%	0.15
Decrease in physical strength	77.14%	60.23%	0.076
Decrease in stamina	77.14%	61.80%	0.104
Feeling a lack of energy	79.41%	61.80%	0.064
Dry skin	73.53%	63.33%	0.285
Weight gain	79.41%	50.00%	0.003
Increased facial hair	37.14%	20.88%	0.06
Changes in appearance, texture or tone of skin	79.59%	50.00%	0.04
Feeling bloated	65.71%	43.96%	0.029
Low backache	73.53%	54.55%	0.055
Frequent urination	51.43%	34.44%	0.081
Involuntary urination when laughing or coughing	42.86%	29.21%	0.146
**Sexual symptoms**			
Change in sexual desire	58.82%	33.70%	0.011
Vaginal dryness during intimacy	37.14%	31.52%	0.547
Avoiding intimacy	45.71%	49.43%	0.711

**Table 3 ijerph-20-06744-t003:** Association between socio-demographics and work-related variables and scores of MENQOL specific dimensions.

Variables	Vasomotor		Psychological		Physical		Sexual	
	Physicians	Nurses	Physicians	Nurses	Physicians	Nurses	Physicians	Nurses
Age								
40–49	1.89 ± 1.66	2.24 ± 2.09	2.83 ± 1.23	2.68 ± 1.98	3.5 ± 1.14	3.18 ± 1.79	2.3 ± 1.64	2.46 ± 2.11
50–54	2.39 ± 1.62	3.51 ± 2.27	3.06 ± 1.38	3.02 ± 2.43	3.46 ± 1.45	3.48 ± 1.97	2.97 ± 1.72	3.05 ± 2.41
55+	3.06 ± 2.08	2.12 ± 1.9	2.29 ± 1.34	2.36 ± 2.01	3.56 ± 1.47	3.23 ± 1.61	3.08 ± 2.32	3.07 ±2.44
*p*-value	0.281 *	0.021 *	0.274 *	0.497 *	0.996 *	0.811 *	0.572 *	0.356 *
Marital status, N (%)								
Married	2.62 ± 1.99	2.45 ± 2.28	2.66 ± 1.22	2.33 ± 2.12	3.29 ± 1.38	3.23 ± 2.01	2.76 ± 1.75	2.72 ± 2.35
Other (Single, Divorced, Widowed)	2.03 ± 1.31	2.76 ± 2.12	2.87 ± 1.56	3.05 ± 2.09	3.91 ± 1.15	3.34 ± 1.66	2.86 ± 2.34	2.79 ± 2.21
*p*-value	0.619 **	0.482	0.619 **	0.125	0.207 **	0.785	0.912 **	0.881
Education, N (%)								
Higher education	2.56 ± 1.93	2.18 ± 1.65	2.81 ± 1.38	2.21 ± 1.38	3.63 ± 1.31	2.96 ± 1.58	3.06 ± 1.98	2.67 ± 1.96
Specialized secondary education	2.67 ± 0	2.82 ± 2.38	2.71 ± 0	2.92 ± 2.32	1.94 ± 0	3.49 ± 1.91	1 ± 0	2.81 ± 2.39
Other *	1.67 ± 0.94	2.33 ± 1.33	2.31 ± 1.18	3.14 ± 3	3.14 ± 1.4	2.46 ± 1.34	1.67 ± 0.82	2.44 ± 2.5
*p*-value	0.626	0.692 *	0.69 *	0.928	0.365 *	0.468 *	0.251 *	0.872 *
Medical organization, N (%)								
National Scientific Center for Mother and Child	3.42 ± 2.27	2.72 ± 2.25	2.85 ± 1.35	2.79 ± 2.16	3.49 ± 1.21	3.47 ± 1.75	2.92 ± 2.17	2.79 ± 2.25
Republican Diagnostic Center	2.21 ± 1.39	2.47 ± 2.03	2.88 ± 1.32	2.56 ± 1.98	3.63 ± 1.41	3.21 ± 1.73	3.14 ± 1.79	2.69 ± 2.33
Republican Children’s Rehabilitation Center	1.67 ± 0.94	4.11 ± 3.56	2.64 ± 1.63	3.33 ± 4.04	3.97 ± 1.28	3.17 ± 3.16	2.5 ± 1.75	3.44 ± 3.15
University Health Center	1 ± 0	2.38 ± 2.02	1.86 ± 1.21	3.53 ± 2.35	2.59 ± 2.25	3.77 ± 2.12	1 ± 0	2.95 ±2.45
*p*-value	0.196 *	0.798	0.702 *	0.745 *	0.643 *	0.85 *	0.323 *	0.936 *
Department, N (%)								
Surgical	3 ± 3.46	2.55 ± 1.91	3.86 ± 0.2	3.14 ± 2.28	4.29 ± 1.61	3.94 ± 1.76	3.78 ± 2.83	3.04 ± 2.53
Internal Medicine	2.87 ± 1.94	3.49 ± 2.72	2.87 ± 1.4	3.41 ± 2.49	3.51 ± 1.16	3.63 ± 2.04	2.98 ± 1.92	3.43 ± 2.22
Other	1.93 ± 0.85	2.37 ± 1.79	2.62 ± 1.31	2.47 ± 1.58	3.35 ± 1.57	2.92 ± 1.47	2.33 ± 1.76	2.42 ± 2.08
*p*-value	0.641 *	0.193	0.508 *	0.374	0.733 *	0.187	0.698 *	0.369
Work hours per week, N (%)								
<40 h	1.33 ± 0.56	3.68 ± 2.47	1.86 ± 1.16	2.93 ± 1.98	2.27 ± 1.58	3.55 ± 1.67	2.22 ± 1.71	3.15 ± 2.27
40–60 h	2.56 ± 1.8	2.14 ± 1.82	2.96 ± 1.31	2.55 ± 2.23	3.79 ± 1.13	3.18 ± 1.95	2.95 ± 1.89	2.73 ± 2.24
>60 h	3.56 ± 2.87	2.55 ± 2.27	2.67 ± 1.37	2.87 ± 2.12	3.67 ± 1.34	3.18 ± 1.69	2.67 ± 2.89	2.26 ± 2.32
*p*-value	0.216 *	0.016	0.167 *	0.745	0.125 *	0.703	0.583 *	0.437
Having night shifts, N (%)								
Yes	2.13 ± 1.76	2.61 ± 2.27	3.11 ± 1.61	2.63 ± 2.04	2.95 ± 0.89	3.23 ± 1.71	3.47 ± 2.82	2.64 ± 2.29
No	2.48 ± 1.84	2.71 ± 2.12	2.68 ± 1.3	2.87 ± 2.23	3.6 ± 1.37	3.41 ± 1.89	2.68 ± 1.74	2.93 ± 2.25
*p*-value	0.523 **	0.829	0.525 **	0.62	0.189 **	0.656	0.567 **	0.567
Have management responsibilities, N (%)								
Yes	3.35 ± 2.14	2.44 ± 1.84	3.26 ± 1.28	2.79 ± 2.02	3.86 ± 1.26	3.23 ± 1.88	3.39 ± 1.93	2.98 ± 2.08
No	1.69 ± 0.89	2.75 ± 2.32	2.29 ± 1.19	2.73 ± 2.17	3.29 ± 1.37	3.38 ± 1.79	1.83 ± 1.28	2.71 ± 2.34
*p*-value	0.019 **	0.589	0.045 **	0.913	0.309 **	0.743	0.015 **	0.636
General satisfaction with job, N (%)								
Satisfied	2.67 ± 1.95	2.39 ± 2.06	2.84 ± 1.27	2.23 ± 1.75	3.67 ± 1.36	2.92 ± 1.57	2.95 ± 1.91	2.37 ± 2.0
Dissatisfied	1.76 ± 0.98	3.16 ± 2.39	2.33 ± 1.52	3.69 ± 2.41	2.98 ± 1.12	4.0 ± 1.97	2.29 ± 1.91	3.5 ± 2.56
*p*-value	0.3 **	0.116	0.289 **	0.002	0.161 **	0.009	0.492 **	0.027
Satisfaction with salary, N (%)								
Satisfied	2.51 ± 1.86	2.56 ± 2.09	2.66 ± 1.24	2.09 ± 1.49	3.79 ± 1.56	2.65 ± 1.36	2.83 ± 1.89	2.46 ± 2.16
Dissatisfied	2.37 ± 1.81	2.68 ± 2.25	2.78 ± 1.41	3.04 ± 2.31	3.31 ± 1.13	3.61 ± 1.91	2.77 ± 1.96	2.92 ± 2.32
*p*-value	0.799 **	0.811	0.715 **	0.059	0.294 **	0.033	0.943 **	0.403
Satisfaction with work environment, N (%)								
Satisfied	2.68 ± 1.99	2.46 ± 2.05	2.95 ± 1.27	2.44 ± 1.93	3.59 ± 1.35	3.09 ± 1.71	2.97 ± 1.9	2.49 ± 2.09
Dissatisfied	1.74 ± 0.91	3.35 ± 2.66	2.05 ± 1.33	3.87 ± 2.49	3.26 ± 1.28	4.13 ± 2.06	2.29 ± 1.93	3.98 ± 2.68
*p*-value	0.242 **	0.259 **	0.064 **	0.012	0.482 **	0.069	0.459 **	0.071 **
Self-reported menopause status, N (%)								
Premenopause	2.14 ± 2.08	2.02 ± 1.62	2.66 ± 1.06	2.53 ± 1.93	3.53 ± 0.94	3.05 ± 1.7	2.52 ± 1.67	2.09 ± 1.82
Perimenopause	2.56 ± 1.5	3.02 ± 2.22	3.31 ± 1.95	2.24 ± 1.24	3.16 ± 1.72	2.8 ± 1.16	3 ± 1.98	2.25 ± 1.35
Postmenopause	2.6 ± 1.77	3.28 ± 2.52	2.56 ± 1.24	3.26 ± 2.46	3.61 ± 1.44	3.88 ± 1.93	2.9 ±2.1	3.89 ± 2.61
*p*-value	0.441 *	0.066 *	0.618 *	0.669 *	0.966 *	0.115 *	0.787 *	0.012 *

*p*-value is calculated using * Kruskal–Wallis test or ** Wilcoxon Rank–Sum test.

## Data Availability

The data will be provided on reasonable request.

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
