# Peer review of "Unveiling the Role of the Work Environment in the Quality of Life of Menopausal Physicians and Nurses"

_ijerph, 2023, doi:10.3390/ijerph20186744_

Round 1
Reviewer 1 Report
This study aims to shed light on an understudied aspect, the frequency of postmenopausal symptoms in women engaged in health care. To correctly frame the value of the contribution, the authors should give more details on the method used.
1. The population. Many readers know little about healthcare in Kazakhstan. Official data indicate that there are over 70,000 doctors. How many women are there? And among the nurses?
2. Sample selection. How were the respondents recruited? Were there differences in participation between hospitals?
3. The sample is divided into premenopause, perimenopause and postmenopause. A definition of the terms would be very helpful; in particular, how to differentiate premenopause from perimenopause?
4. The questionnaire. A review of 25 different translations identified many modifications to the original questionnaire, including omission or addition of items and alterations to the validated methodological analysis [Sydora BC, Fast H, Campbell S, Yuksel N, Lewis JE, Ross S. Use of the Menopause-Specific Quality of Life (MENQOL) questionnaire in research and clinical practice: a comprehensive scoping review. Menopause. 2016 Sep;23(9):1038-51. doi: 10.1097/GME.0000000000000636.]. The Authors should report on the features of the version used and should discuss this issue in the Limitations section.
5. The authors did not analyze the variation of symptoms as a function of age, nor the distance from menopause.
6. The study focused on the difference in work but did not consider numerous other factors that are correlated with the syndrome, such as physical activity, diet, and others. An example: Barati M, Akbari-Heidari H, Samadi-Yaghin E, Jenabi E, Jormand H, Kamyari N. The factors associated with the quality of life among postmenopausal women. BMC Womens Health. 2021 May 18;21(1):208. doi: 10.1186/s12905-021-01361-x.
Author Response
Reviewer 1:
This study aims to shed light on an understudied aspect, the frequency of postmenopausal symptoms in women engaged in health care. To correctly frame the value of the contribution, the authors should give more details on the method used.
Response – Dear reviewer, thank you for your valuable comments. We have addressed each comment and in the text of the revised manuscript we highlighted all changes in yellow.
Comment #1: The population. Many readers know little about healthcare in Kazakhstan. Official data indicate that there are over 70,000 doctors. How many women are there? And among the nurses?
Response – Thank you for the comment. Additional details about the country and the healthcare were included into the introduction part text:
“Kazakhstan is one of the Central Asian countries. After the Soviet Union collapse and subsequent economic recession, the Kazakhstani healthcare system passed through pro-found reforms. Notably, only around 3% of gross domestic product (GDP) is allocated for the healthcare system, while the average expenditures in the post-Soviet neighboring countries are around 6.5% [9,10]. Government priorities in this sector include developing the country’s primary healthcare networks, improving its public health administration system, providing expanded medical personnel training, enhancing mother and child health services, and emphasizing preventive measures such as diagnostics, treatment of lifestyle diseases, patient rehabilitation and development of personalized medicine. The Kazakhstani government has approved a budget of USD 7.5 billion for the State Program for the Development of Healthcare 2020-2025 [11]. According to the National Agency for Statistics, as of January 1, 2023, more than 260,200 medical workers, including 76,000 doctors, and 186,000 nurses, work in the Kazakhstani healthcare system [12].”
Comment #2: Sample selection. How were the respondents recruited? Were there differences in participation between hospitals?
Response – The relevant clarification was added in methods part:
“An email containing an invitation to participate in the study, the informed consent form, and a questionnaire was sent to the participants' corporate email addresses. Participation was entirely voluntary, and individuals had the freedom to discontinue the survey when-ever they chose. Those who agreed to take part returned the completed questionnaire and a signed informed consent form to the researchers.”
Comment #3: The sample is divided into premenopause, perimenopause and postmenopause. A definition of the terms would be very helpful; in particular, how to differentiate premenopause from perimenopause?
Response – The following information concerning this comment was added in the methods part:
“Premenopause – a period in women’s life preceding the onset of menopause”
Comment #4: The questionnaire. A review of 25 different translations identified many modifications to the original questionnaire, including omission or addition of items and alterations to the validated methodological analysis [Sydora BC, Fast H, Campbell S, Yuksel N, Lewis JE, Ross S. Use of the Menopause-Specific Quality of Life (MENQOL) questionnaire in research and clinical practice: a comprehensive scoping review. Menopause. 2016 Sep;23(9):1038-51. doi: 10.1097/GME.0000000000000636.]. The Authors should report on the features of the version used and should discuss this issue in the Limitations section.
Response – We appreciate the valuable comment. As an response we elaborated this issue in the methods part:
“The questionnaire underwent cross-cultural adaptation and translation for use in the study settings. The translation process involved professional trilingual (English-Russian-Kazakh) interpreters conducting both forward and backward translations into Kazakh and Russian languages. Discrepancies in translations and interpretations were resolved through comprehensive discussions by a panel of experts. The Kazakh and Russian translated versions of the survey were pretested on 20 participants to evaluate the clarity and appropriateness of the translated items and responses. In this study, the Cronbach α values were 0.91 for vasomotor domain, 0.95 for psychological domain, 0.95 for physical domain and 0.9 for sexual domain.”
Comment #5: The authors did not analyze the variation of symptoms as a function of age, nor the distance from menopause.
Response – Thank you for this comment. We initially did not intended to analyze the variation of symptoms by dividing the sample based on their age. However, we agree that this might be an interesting analysis for readers. Thus, we added the following supplement table 1, which can be found in the revised manuscript.
Comment #6: The study focused on the difference in work but did not consider numerous other factors that are correlated with the syndrome, such as physical activity, diet, and others. An example: Barati M, Akbari-Heidari H, Samadi-Yaghin E, Jenabi E, Jormand H, Kamyari N. The factors associated with the quality of life among postmenopausal women. BMC Womens Health. 2021 May 18;21(1):208. doi: 10.1186/s12905-021-01361-x.
Response – We appreciate the insightful comment of the reviewer. We initially aimed to investigate the associations between characteristics of work environment and Quality of life of physicians and nurses. Therefore, unfortunately, our questionnaire did not include the items evaluating the physical activity, diet and other important determinants.
Reviewer 2 Report
I have carefully read the manuscript entitled: “Unveiling the Role of the Work Environment in the Quality of Life of Menopausal Physicians and Nurses.” In it, the authors explore through the use of instruments the relationship between work environment and the menopausal quality of life (QoL) in health professionals.
The work presented by the authors addresses a relevant topic. It is of interest to the health area because it focuses on a study population that requires care and understanding regarding an organic condition such as menopause.
Below I point out my comments related to this investigation. These are shown concerning their order of appearance in the manuscript.
Remove the parentheses that are in the abstract. Lines 23, 26, 28, and 35.
Line 29. Indicate the statistical parameter to indicate that the study groups' differences were statistically significant.
In the introduction, the authors adequately describe the relevance of their study and the importance of carrying it out with the referred health personnel.
Line 87. Separate value by study groups.
Line 88. Define the number of hospitals.
Lines 91 and 92. The authors must indicate the sampling strategy (calculation of sample size and type of sampling). They must also indicate the percentage of those who did not wish to participate.
Lines 99 and 100. Before "Physicians and nurses who took part in the study were required to complete a self-reported questionnaire" mention that a questionnaire was applied to explore sociodemographic and labor variables.
In the results section, it is recommended to use asterisks in those values that are statistically significant and some other symbol in the cases that the authors consider relevant or informative. It needs to be clarified that the asterisks are sometimes in the text of the tables and other times in the values since there is no explanation at the bottom.
Lines 224-239. It is recommended to omit or synthesize this information because it is previously described in the results section.
It is recommended to strengthen the discussion section with more references and address the importance of these studies in risk groups.
In the limitations section, the authors should be more emphatic that due to the nature of the study, the type of sampling, the total sample value (n), and of the subgroups analyzed, the research presents little representativeness and external validity.
Lines 308-310. Delete this sentence because the title of the work was already described.
Minor languages changes are required.
Author Response
Reviewer 2:
I have carefully read the manuscript entitled: “Unveiling the Role of the Work Environment in the Quality of Life of Menopausal Physicians and Nurses.” In it, the authors explore through the use of instruments the relationship between work environment and the menopausal quality of life (QoL) in health professionals.
Response – Dear reviewer, thank you for your valuable comments. We have addressed each comment and in the text of the revised manuscript we highlighted all changes in yellow.
The work presented by the authors addresses a relevant topic. It is of interest to the health area because it focuses on a study population that requires care and understanding regarding an organic condition such as menopause.
Response – We appreciate the reviewer’s comment.
Below I point out my comments related to this investigation. These are shown concerning their order of appearance in the manuscript.
Comment #1: Remove the parentheses that are in the abstract. Lines 23, 26, 28, and 35.
Response – We introduced changes based on reviewer’s suggestion.
Comment #2: Line 29. Indicate the statistical parameter to indicate that the study groups' differences were statistically significant.
Response – We introduced changes based on reviewer’s suggestion.
Comment #3: In the introduction, the authors adequately describe the relevance of their study and the importance of carrying it out with the referred health personnel.
Comment #4: Line 87. Separate value by study groups.
Response – We introduced changes based on reviewer’s suggestion.
Comment #5: Line 88. Define the number of hospitals.
Response – We introduced changes based on reviewer’s suggestion.
Comment #6: Lines 91 and 92. The authors must indicate the sampling strategy (calculation of sample size and type of sampling). They must also indicate the percentage of those who did not wish to participate.
Response – We thank reviewer for the valuable comment. We clarified the sampling strategy in the methods part:
“The required sample size was calculated based on an anticipated effect size, a Type I error rate of 5%, and a power of 80%. The sample size was estimated to be 240 participants. However, due to non-response, we were able to obtain 130 eligible responses. Thus, the response rate accounted for 54%.”
Comment #7: Lines 99 and 100. Before "Physicians and nurses who took part in the study were required to complete a self-reported questionnaire" mention that a questionnaire was applied to explore sociodemographic and labor variables.
Response – We introduced changes based on reviewer’s suggestion.
Comment #8: In the results section, it is recommended to use asterisks in those values that are statistically significant and some other symbol in the cases that the authors consider relevant or informative. It needs to be clarified that the asterisks are sometimes in the text of the tables and other times in the values since there is no explanation at the bottom.
Response – Thank you for the comment. We introduced changes based on reviewer’s suggestion.
Comment #9: Lines 224-239. It is recommended to omit or synthesize this information because it is previously described in the results section.
Response –The discussion part was restructured and previous information was synthesized.
Comment #10: It is recommended to strengthen the discussion section with more references and address the importance of these studies in risk groups.
Response – More detailed analysis of previous findings on this topic was made to strengthen the discussion session.
Comment #11: In the limitations section, the authors should be more emphatic that due to the nature of the study, the type of sampling, the total sample value (n), and of the subgroups analyzed, the research presents little representativeness and external validity.
Response – Thank you for the comment. We elaborated on the limitations of the study.
Comment #12: Lines 308-310. Delete this sentence because the title of the work was already described.
Response – We introduced changes based on reviewer’s suggestion.
Reviewer 3 Report
Thank you very much for giving me the opportunity to review the manuscript entitled “Unveiling the Role of the Work Environment in the Quality of Life of Menopausal Physicians and Nurses”. The quality of life of menopausal women and the search for factors that favor or hinder coping with bothersome menopausal symptoms are issues important from the point of view of public health and within the area of interest of IJERPH.
However the study has important limitations.
1. The authors did not formulate any hypothesis that they intended to verify when planning their research. The reasons for the differences in the quality of life and the symptoms of menopause should not be sought in the name of the profession (nurse - doctor), but in the characteristics of physical and mental workload and the conditions in which work is performed.
2. The authors write that the aim of the study was to determine the frequency of menopausal symptoms and to assess the quality of life among doctors and nurses in Kazakhstan. They found that while there were no differences in overall MENQoL or their individual domains between physicians and nurses, physicians reported having menopausal symptoms more frequently than nurses, and nurses reported no menopausal symptoms more frequently than physicians. At the same time, several factors were identified that significantly contributed to the MENQoL scores specific to physicians and nurses. Among nurses, age and working hours per week were associated with vasomotor symptoms, and among physicians, performing managerial duties was associated with higher levels of anxiety about vasomotor, psychological, and sexual symptoms of menopause. In addition, it was found that satisfaction with the overall work and its individual aspects significantly influenced the assessment of the quality of life of nurses, and the menopausal status was significantly related to the sexual aspect of the quality of life of nurses. Unfortunately, apart from stating the actual state of affairs, nothing from the conducted research shows. The authors did not try to make any summary or generalization, and in the discussion and conclusions they only repeat the results of their own research. All this means that the submitted manuscript is more of a research report than a scientific article.
A minor comment:
1. The article by D'Angelo et al (2022) appears twice in the reference list as No. 9 and No. 17.
2. Tables should be prepared in such a way that they are understandable regardless of the text of the article. All abbreviations and distinctions (markings) in the tables should be described below the table. I couldn't find an explanation of what the asterisk (*) in tables 1-3 means.
Author Response
Reviewer 3:
Thank you very much for giving me the opportunity to review the manuscript entitled “Unveiling the Role of the Work Environment in the Quality of Life of Menopausal Physicians and Nurses”. The quality of life of menopausal women and the search for factors that favor or hinder coping with bothersome menopausal symptoms are issues important from the point of view of public health and within the area of interest of IJERPH.
Response - Dear reviewer, thank you for your valuable comments. We have addressed each comment and in the text of the revised manuscript we highlighted all changes in yellow.
However the study has important limitations.
Comment #1: The authors did not formulate any hypothesis that they intended to verify when planning their research. The reasons for the differences in the quality of life and the symptoms of menopause should not be sought in the name of the profession (nurse - doctor), but in the characteristics of physical and mental workload and the conditions in which work is performed.
Response – Thank you for the comment. We added the following information in the introduction part of the manuscript:
“We hypothesize that physicians and nurses possess varying QoL scores due to differences in their associated work-related characteristics.”
Comment #2: The authors write that the aim of the study was to determine the frequency of menopausal symptoms and to assess the quality of life among doctors and nurses in Kazakhstan. They found that while there were no differences in overall MENQoL or their individual domains between physicians and nurses, physicians reported having menopausal symptoms more frequently than nurses, and nurses reported no menopausal symptoms more frequently than physicians. At the same time, several factors were identified that significantly contributed to the MENQoL scores specific to physicians and nurses. Among nurses, age and working hours per week were associated with vasomotor symptoms, and among physicians, performing managerial duties was associated with higher levels of anxiety about vasomotor, psychological, and sexual symptoms of menopause. In addition, it was found that satisfaction with the overall work and its individual aspects significantly influenced the assessment of the quality of life of nurses, and the menopausal status was significantly related to the sexual aspect of the quality of life of nurses. Unfortunately, apart from stating the actual state of affairs, nothing from the conducted research shows. The authors did not try to make any summary or generalization, and in the discussion and conclusions they only repeat the results of their own research. All this means that the submitted manuscript is more of a research report than a scientific article.
Response – Significant changes were made in discussion and conclusion parts with more detailed analysis and synthesis of previous studies. The general conclusions and suggestions from our study results were included.
A minor comment:
Comment #3: The article by D'Angelo et al (2022) appears twice in the reference list as No. 9 and No. 17.
Response – We revised all the references and corrected the error that was pointed out by the reviewer.
Comment #4: Tables should be prepared in such a way that they are understandable regardless of the text of the article. All abbreviations and distinctions (markings) in the tables should be described below the table. I couldn't find an explanation of what the asterisk (*) in tables 1-3 means.
Response – We introduced changes based on reviewer’s suggestion.
Round 2
Reviewer 1 Report
The authors conveniently addressed the comments
Reviewer 3 Report
In my opinion the manuscript has been sufficiently improved to warrant publication in IJERPH.